# The Secretome of Irradiated Peripheral Mononuclear Cells Attenuates Hypertrophic Skin Scarring

**DOI:** 10.3390/pharmaceutics15041065

**Published:** 2023-03-25

**Authors:** Vera Vorstandlechner, Dragan Copic, Katharina Klas, Martin Direder, Bahar Golabi, Christine Radtke, Hendrik J. Ankersmit, Michael Mildner

**Affiliations:** 1Laboratory for Cardiac and Thoracic Diagnosis, Regeneration and Applied Immunology, Department of Thoracic Surgery, Medical University of Vienna, 1090 Vienna, Austria; 2Aposcience AG, 1200 Vienna, Austria; 3Department of Plastic and Reconstructive Surgery, Medical University of Vienna, 1090 Vienna, Austria; 4Department of Medicine III, Division of Nephrology and Dialysis, Medical University of Vienna, 1090 Vienna, Austria; 5Department of Orthopedics and Trauma-Surgery, Medical University of Vienna, 1090 Vienna, Austria; 6Department of Dermatology, Medical University of Vienna, 1090 Vienna, Austria

**Keywords:** scar, regeneration, peripheral blood mononuclear cell secretome

## Abstract

Hypertrophic scars can cause pain, movement restrictions, and reduction in the quality of life. Despite numerous options to treat hypertrophic scarring, efficient therapies are still scarce, and cellular mechanisms are not well understood. Factors secreted by peripheral blood mononuclear cells (PBMCsec) have been previously described for their beneficial effects on tissue regeneration. In this study, we investigated the effects of PBMCsec on skin scarring in mouse models and human scar explant cultures at single-cell resolution (scRNAseq). Mouse wounds and scars, and human mature scars were treated with PBMCsec intradermally and topically. The topical and intradermal application of PBMCsec regulated the expression of various genes involved in pro-fibrotic processes and tissue remodeling. We identified elastin as a common linchpin of anti-fibrotic action in both mouse and human scars. In vitro, we found that PBMCsec prevents TGFβ-mediated myofibroblast differentiation and attenuates abundant elastin expression with non-canonical signaling inhibition. Furthermore, the TGFβ-induced breakdown of elastic fibers was strongly inhibited by the addition of PBMCsec. In conclusion, we conducted an extensive study with multiple experimental approaches and ample scRNAseq data demonstrating the anti-fibrotic effect of PBMCsec on cutaneous scars in mouse and human experimental settings. These findings point at PBMCsec as a novel therapeutic option to treat skin scarring.

## 1. Introduction

Skin scarring after surgery, trauma, or burn injury is a major problem affecting 100 million people every year, causing a significant global disease burden [1]. Patients with hypertrophic scars, occurring in 40–90% of cases after injury [2], suffer from pain, pruritus, and reduced quality of life [3,4]. Skin scarring has been extensively studied [5,6], and recently, we were able to elucidate hypertrophic scar formation at the single-cell level [7]. However, many cellular mechanisms remain unclear, and for most conservative therapeutic options, we have low evidence of their efficacy [8]. Wound healing and scar formation are complex, rigidly coordinated processes, with multiple cell types being involved [9]. Wound healing is characterized by an acute inflammatory phase, a proliferative phase, and a remodeling phase [9]. Prolonged inflammation results in increased fibroblast (FB) activity, with enhanced secretion of transforming growth factor beta 1 (TGFβ1), TGFβ2, insulin-like growth factor (IGF1), and other cytokines [10,11]. TGFβ1 induces the differentiation of FBs into myofibroblasts (myoFBs) [12]. myoFBs show strong contractility and excessively deposit extracellular matrix (ECM) components, eventually leading to (hypertrophic) scar formation. Matured (hypertrophic) scars show dense, parallel ECM and strong tissue contraction [12].

Numerous pharmaceutical attempts to tackle hypertrophic scars have been proposed during recent decades, e.g., intralesional injection of corticosteroids, 5-Fluorouracil (5-FU), or triamcinolone (TAC) [13,14]. Other therapeutic options include compression therapy or topical silicone application. These therapies, however, still lack evidence of efficacy and safety and show high recurrence rates, and the mechanisms of action are not well understood [15,16]. In recent years, numerous pre-clinical studies have shown effective scar treatment or improvement in scar formation after the application of conditioned media derived from different stem cell populations, such as amniotic mesenchymal stem cells (MSCs) [17], fat-derived stem cells [18], bone marrow-induced MSCs [19], and induced pluripotent stem cells [20], amongst others [21]. However, the transferability of promising pre-clinical animal studies to humans was shown to be limited [22]. Furthermore, autologous conditioned media from various stem cell populations have significant disadvantages, as the production of these secretomes is expensive and hardly scalable, due to the limited numbers of available stem cells [23].

Hence, the idea of cell-free paracrine therapies in an allogeneic setting has drawn increasing attention. As different kinds of stem cells still have the same limitations in the allogeneic setting, peripheral blood mononuclear cells (PBMCs) have been proposed as an alternative source of paracrine factors [24].

The secretome of irradiated peripheral blood mononuclear cells (PBMCsec) has been extensively studied in recent years, showing encouraging pre-clinical results. PBMCsec has been found to enhance wound healing [25,26,27], elicit angiogenic effects [26,28], prevent platelet aggregation and vasodilation [29], exert anti-microbial activity [30], attenuate neurological damage in focal ischemia [31] and spinal cord injury [32], and regenerate infarcted myocardium [33]. Moreover, PBMCsec has been shown to reduce the activation of mast cells and basophils [34] and reduce the maturation and antigen uptake of dendritic cells, as well as dendritic cell-mediated T-cell priming [35]. In a phase I study, PBMCsec was found to be safe and well tolerated in the topical application of autologous PBMCsec on skin wounds [36]. In addition, a phase II clinical trial on the efficacy of allogeneic PBMCsec in patients with diabetic foot ulcers is currently ongoing [37]. It should be noted that the favorable pleiotropic effects of PBMCsec cannot be broken down to a single mode of action [37], as PBMCsec has repeatedly demonstrated its regenerative power with the synergistic action of all components, including proteins, lipids, extracellular vesicles, and nucleic acids [26,28,37].

Therefore, we attempt to provide a multi-model murine and human approach at the single-cell level to identify the potential mechanisms of action of PBMCsec on skin scarring. Due to the plethora of beneficial effects of PBMCsec, we hypothesized that PBMCsec prevents (hypertrophic) scarring or improves tissue quality in already persisting scars. In this study, we demonstrate the anti-fibrotic activity of PBMCsec and provide mechanistic insights into its anti-fibrotic effect. This study facilitates the investigation of PBMCsec for its future clinical use as a treatment option for skin scarring.

## 2. Materials and Methods

### 2.1. Ethics Statement

The use of healthy abdominal skin (Vote No. 217/2010) and scar tissue (Vote No. 1533/2017) was approved by the ethics committee of Medical University of Vienna. Animal experiments were approved by the ethics committee of Medical University of Vienna and the Austrian Federal Ministry of Education, Science, and Research (Vote No. BMBWF-66.009/0075-V/3b/2018).

### 2.2. Patient Material

Resected scar tissue was obtained from three patients who underwent elective scar resection surgery after giving informed consent. Scars were previously classified as hypertrophic, pathological scars according to the Patient and Observer Scar Assessment Scale (POSAS) [38] by a plastic surgeon. All scars were mature scars, i.e., they were at least two years old; had not been operated on; and had not been previously treated with corticosteroids, 5-FU, irradiation, or similar treatments. All scar samples were obtained from male and female patients younger than 45 years old, with no chronic diseases nor chronic medication. Healthy skin was obtained from three healthy female donors between 25 and 45 years of age from surplus abdominal skin removed during elective abdominoplasty.

### 2.3. Animals

In all mouse experiments, 8–12-week-old female Balb/c mice (Medical University of Vienna Animal Breeding Facility, Himberg, Austria) were used. Mice were housed in a selected pathogen-free environment according to enhanced standard husbandry with a 12/12 h light/dark cycle and ad libitum access to food and water.

### 2.4. Full-Thickness Wound and Scarring Model in Mice

For the full-thickness skin wound and scarring model, mice were deeply anesthetized with ketamine 80–100 mg/kg and xylazine 10–12.5 mg/kg i.p. They were given postoperative analgesia with the s.c. injection of 0.1 mL/10 mg Buprenorphin and 7.5 mg/mL Piritramid in drinking water. A 9 × 9 mm square area was marked on the back and excised with sharp scissors. The wounds were left to heal uncovered without any further intervention for 4 weeks, and the resulting scar tissue was observed and photographed.

### 2.5. Production of Irradiated Mononuclear Cell Secretome (PBMCsec)

The secretome of human PBMCs was produced in compliance with good manufacturing practice (GMP) by the Austrian Red Cross, Blood Transfusion Service for Upper Austria (Linz, Austria), as previously described [26,39] (Appendix A). PBMCs were obtained with Ficoll-Paque PLUS (GE Healthcare, Chicago, IL, USA)-assisted density gradient centrifugation, adjusted to a concentration of 25 × 106 cells/mL (25 U/mL; 1 Unit = secretome of 1 million cells) and exposed to 60 Gy cesium 137 gamma irradiation (IBL 437C; Isotopen Diagnostik CIS GmbH, Dreieich, Germany). Cells were cultured in phenol red-free CellGenix GMP DC medium (CellGenix GmbH, Freiburg, Germany) for 24 ± 2 h. Cells and cellular debris were removed with centrifugation, and supernatants were passed through a 0.2 µm filter. Methylene blue treatment was performed as described [40] for viral clearance. The secretome was lyophilized, terminally sterilized with high-dose gamma irradiation, and stored at −80 °C. All experiments were performed using secretomes of the following batches produced under GMP: A000918399086, A000918399095, A000918399098, A000918399101, A000918399102, and A000918399105. Immediately before performing the experiments, the lyophilizate was reconstituted in 0.9% NaCl to the original concentration of 25 U/mL.

### 2.6. PBMCsec Injection into Mouse Scars

Starting on day 29 after skin wounding, mice were injected with 100 µL 0.9% NaCl, medium (phenol red-free CellGenix GMP DC medium), or PBMCsec, which was prepared as described above, every second day for two weeks. Subsequently, half of the mice from each group (n = 2) were sacrificed and analyzed, while the other half (n = 2) were left for another two weeks without further intervention and then sacrificed.

### 2.7. PBMCsec Topical Application on Mouse Scars

Starting on the day of skin wounding (d0), mouse scars were treated with PBMCsec, medium, or NaCl 0.9%. Ultrasicc/Ultrabas ointment (1:1; Hecht-Pharma, Bremervörde, Germany) was used as a carrier substance for all treatments. Four parts of Ultrasicc/Ultrabas (50:50) and one part of water were mixed and used as control treatment (i.e., 100 µL contained 40 µL of Ultrasicc, 40 µL of Ultrabas, and 20 µL of agent or control). Then, 5 U/mL (200 µL of dissolved lyophilizate) PBMCsec or 200 µL/mL medium was mixed with ointment. Mice were treated with control or inhibitors by applying 100 µL of ointment on each wound immediately after wounding.

After application, mice were individually placed in empty cages without litter for 30 min and closely monitored to prevent immediate removal of the treatments and achieve sufficient tissue resorption. Scabs were left intact to prevent wound infections. Mice were treated daily for the first 7 days and thrice a week for 7 weeks. After scar formation, 4 mm biopsies of the scar tissue were taken and cut in half. One half of each scar sample was used for histological analysis, and the other biopsy halves from each treatment group were pooled and analyzed together with scRNAseq as described below.

### 2.8. Ex Vivo Skin and Scar Stimulation

From human skin and scar tissue, 6 mm punch biopsies were taken; subcutaneous adipose tissue was removed; and biopsies were placed in 12-well plates supplemented with 400 µL of DMEM (Gibco, Thermo Fisher, Waltham, MA, USA; with 10% fetal bovine serum and 1% penicillin/streptomycin) and 100 µL of CellGenix medium or 100 µL of PBMCsec. In addition, 100 mL of medium or PBMCsec was injected into the upper dermis in the middle of the biopsy. Biopsies were incubated for 24 h and then harvested for scRNAseq analysis. Sample “Skin 1 medium” was lost due to technical difficulties during preparation.

### 2.9. Skin and Scar PBMCsec Stimulation, Cell Isolation, and Droplet-Based scRNAseq

Mouse scars and stimulated human skin and scar samples were digested using Miltenyi Whole Skin dissociation Kit (Miltenyi Biotec, Bergisch-Gladbach, Germany) for 2.5 h according to the manufacturer’s protocol and processed using GentleMACS OctoDissociator (Miltenyi). The cell suspension was filtered through a 100 µm filter and a 40 µm filter, centrifuged for 10 min at 1500 rpm, washed twice, and resuspended in 0.04% FBS in phosphate-buffered saline (PBS). DAPI was added at 1 µL/1 million cells for 30 s; cells were washed twice and sorted for viability using a MoFlo Astrios high-speed cell-sorting device (Beckman-Coulter, Indianapolis, IN, USA). Only distinctly DAPI-negative cells were used for further processing. Immediately after sorting, viable cells were loaded onto a 10X-chromium instrument (Single cell gene expression 3′v 2/3; 10X Genomics, Pleasanton, CA, USA) to generate a gel bead in emulsion (GEM). GEM generation, library preparation, RNA sequencing, demultiplexing, and counting were performed at Biomedical Sequencing Core Facility of Center for Molecular Medicine (CeMM; Vienna, Austria). Sequencing was performed using 2 × 75 bp, paired-end, with Illumina HiSeq 3000/4000 (Illumina, San Diego, CA, USA).

### 2.10. Cell–Gene Matrix Preparation and Downstream Analysis

Raw sequencing reads were demultiplexed and aligned to the human (GrChH38) and mouse (mm10) reference genomes using the Cell Ranger mqfast and count pipelines (v4.0; 10× Genomics, Pleasanton, CA, USA) to generate cell–gene matrices. The cell–gene matrices were then loaded into “Seurat” (v4.0; Satija Lab, New York, USA) in an R environment (v4.1.2; R Foundation for Statistical Computing, Vienna, Austria) and processed according to the recommended standard workflow for the integration of several datasets [41,42]. All human skin and scar samples were integrated in a single integration; likewise, all mouse samples were integrated in a single integration. Cells with less than 500 or more than 4000 detected genes, more than 20,000 reads per cell, or a mitochondrial gene count higher than 5% were removed from the dataset to ensure high data quality. After principal component analysis and the identification of significant principal components using the Jackstraw procedure [43], cells were clustered using non-linear dimensional reduction with uniform manifold approximation and projection (UMAP). Differentially expressed genes were calculated in Seurat using the Wilcoxon rank-sum test with Bonferroni correction.

In all datasets, normalized count numbers were used for differential gene expression analysis and for visualization in violin plots, feature plots, and dot plots, as recommended by the guidelines [44]. In all datasets, cell types were identified according to well-established marker gene expression. To avoid the calculation of batch effects, the normalized count numbers of genes present in the integrated dataset were used to identify differentially expressed genes (DEGs). As keratin and collagen genes were previously found to contaminate skin biopsy datasets and potentially provide a false-positive signal [45], these genes (*COL1A1*, *COL1A2*, and *COL3A1*; *KRT1*, *KRT5*, *KRT10*, *KRT14*, and *KRTDAP*) were excluded from DEG calculation in non-fibroblast clusters (collagens) or non-keratinocyte clusters (keratins), respectively. Moreover, genes *Gm42418*, *Gm17056*, and *Gm26917* caused technical background noise and batch effect in mouse scRNAseq, as previously described [46], and were thus excluded from the dataset.

### 2.11. Gene Ontology (GO) Calculation and Dot Plots

Gene lists of significantly regulated genes (adjusted *p*-value < 0.05; average log fold change (avg_logFC) > 0.1) were inputted into “GO_Biological_Process_2018” in the EnrichR package in R (v3.0; MayanLab, Icahn School of Medicine at Mount Sinai, New York, NY, USA). Dot plots were generated using ggplot2 (H. Wickham. ggplot2: Elegant Graphics for Data Analysis. Springer-Verlag New York, 2016) with color indicating adjusted *p*-value and size showing the odds ratio, sorted by adjusted *p*-value. 

### 2.12. GSEA Matrisome Dot Plots

Curated matrisome gene lists for the terms ”NABA_ECM_GLYCOPROTEINS”, ”NABA_COLLAGENS”, ”NABA_PROTEOGLYCANS”, ”NABA_ECM_REGULATORS”, and ”REACTOME_ELASTIC_FIBRE_FORMATION” were retrieved from the Gene Set Enrichment Analysis platform https://www.gsea-msigdb.org/gsea/index.jsp, accessed on 22 June 2022 [47], and gene names were used to generate dot plots.

### 2.13. TGFβ Injection Fibrosis Model in Mouse Skin

Mice were anesthetized with 3% isoflurane for three minutes. An intrascapular area of approximately 1 × 1 cm area was marked on the skin with a permanent marker. In total, 800 ng of TGFβ1 dissolved in 100 µL of NaCl 0.9%, medium, or PBMCsec (2.5 U) was injected in the marked area for 5 consecutive days, and mice sacrificed on the 6th day. The marked injection areas were biopsied and prepared for histological analysis.

### 2.14. Isolation of Primary Skin FBs

Primary skin and scar FBs were isolated as previously described [7]. In brief, skin or scar samples were incubated overnight in Dipase II (Roche, Basel, Switzerland). Subsequently, the epidermis was removed, and the dermis was incubated in Liberase (Merck Millipore, Burlington, MA, USA) for two hours at 37 °C. Afterwards, the tissue was filtered and rinsed with PBS, and the cells were plated in a T175 cell culture flask and cultured until they reached 90% confluency.

### 2.15. Western Blots

Western blotting was performed as previously described [7]. In brief, after cell lysis in 1× Laemmli buffer, the lysates were separated on SDS-PAGE gels (Bio-Rad Laboratories, Inc., Hercules, CA, USA), and proteins were transferred to nitrocellulose membranes and blocked with non-fat milk. After overnight incubation at 4 °C with the primary antibody (table of antibodies used reported in Appendix A), the membranes were incubated with a horseradish peroxidase-conjugated secondary antibody and imaged.

### 2.16. Immunofluorescence, H&E, and EvG Staining

Immunofluorescence staining on formalin-fixed, paraffin-embedded (FFPE) sections of human and mouse skin and scar tissues was performed according to the protocol provided by the respective antibody manufacturer (table of antibodies used reported in Appendix A) as previously described [7]. Hematoxylin and eosin (H&E) staining and Elastica van Gieson (EvG)-staining were performed at Department of Pathology of Medical University of Vienna according to standardized clinical staining protocols.

### 2.17. TGFβ1-Induced Myofibroblast Differentiation

TGFβ1 stimulation of primary FBs was performed as previously reported [7]. Isolated primary FBs were plated in 6-well plates after the first passage and grown until they reached 100% confluency. FBs were then stimulated with 10 ng/mL TGFβ1 (HEK-293-derived; Peprotech, Rocky Hill, NJ, USA) and with medium or PBMCsec for 24 h. The supernatants were removed, and medium or PBMCsec was resupplied for another 24 h. The supernatants were collected and stored at −80 °C, and cells were lysed in 1× Laemmli buffer (Bio-Rad Laboratories, Inc.) for further analysis.

### 2.18. Elastase Assay

To measure elastase activity, a commercial kit (EnzChek^®^ Elastase Assay Kit; E-12056; Thermo Fisher) was used according to the manufacturer’s instructions. Elastase was applied at 250 mU/mL and incubated with NaCl 0.9% (“Ctrl”), medium, or PBMCsec at 1:1 with assay buffer. Fluorescence intensity was measured with a BMG Fluostar Optima plate reader (BMG Labtech, Ortenberg, Germany) at 505/515 nm wavelength (excitation/emission). Raw values were blank-corrected and normalized to the % of the averaged 4 h of the Ctrl samples. Samples were measured 10 min, 1 h, 2 h, 3 h, and 4 h after elastase application. Statistical analysis was performed with a mixed-effects model for the time factor, with Tukey’s multiple comparisons test.

### 2.19. ELISA

The supernatants of TGFβ1-stimulated FBs after treatment with PBMCsec or controls were collected, centrifuged, and stored at −20 °C for further use. The protein levels of human elastin (ELISA; LS-F4567; LSBio, Seattle, WA, USA) were measured according to the manufacturer’s manual. Absorbance was detected with a FluoStar Optima microplate reader (BMG Labtech).

## 3. Results

### 3.1. PBMCsec Improves Scar Formation in Mice after Topical Treatment during Wound Healing and Intradermal Injection of Preformed Scars

As our previous study on wound healing in pig burn wounds revealed a trend towards better tissue elasticity and less stiffness in early pig burn scars [27], we aimed to investigate the effect of PBMCsec on scar formation and on already existing scars in more detail at the single-cell level.

To achieve this, we created full-thickness excision wounds on the back of 6–8-week-old female Balb/c mice and immediately treated with the topical application of PBMCsec for 8 weeks (Figure 1A). In a separate set of experiments, we allowed the scars to develop for 4 weeks after wounding without further intervention and treated the formed scars with intradermal injection for 2 weeks. Scars were either analyzed right after the two weeks of treatment or after two additional weeks without further treatment to determine whether treatment-associated changes were permanent (Figure 1B).

As previously demonstrated with the secretome of non-irradiated PBMCs [25,26] or PBMCsec in diabetic mice [25,26], we found enhanced wound healing in wild-type mice after the topical application of an emulsion containing PBMCsec (Figure 1C). PBMCsec reduced the wound size significantly more (40 ± 14% of the wound size) than NaCl (72 ± 16) and the control medium alone (60 ± 19%) (Figure 1D). Compared with the intradermal injection of controls, scars appeared softer and reduced in size after the injection of PBMCsec (Figure 1E). Histologically, scars showed a looser structure and reduced fiber density after topical PBMCsec treatment, as evidenced by hematoxylin/eosin (Figure 1F) and Elastica van Gieson (EvG) staining (Figure 1G). Of note, scars treated with intradermal injection exhibited a high number of infiltrating leukocytes (Figure 1H), presumably due to repeated tissue irritation with injections. However, the matrix was looser, and the orientation of collagen fibers showed more vertical structures after the injection of PBMCsec (Figure 1I). These results indicate that PBMCsec not only improves wound healing but also scar formation and the quality of already existing scars in mice.

### 3.2. PBMCsec Induces Significant Changes in the Transcriptome after Topical and Intradermal Application

Next, we performed scRNAseq on scar tissue from the different experimental settings. After quality control (Appendix A) we defined clusters based on well-established marker genes [7,48] from scRNAseq of topically treated scars (Appendix A). Clusters were constantly aligned homogenously under all conditions (Figure 2A,E) and were grouped into fibroblasts (FBs), smooth muscle cells and pericytes (SMCs/PCs), endothelial and lymphatic endothelial cells (ECs), macrophages (Macro), Langerhans cells and dendritic cells (DCs), T cells and B cells (TCs), keratinocytes (KCs), hair follicular cells (HFs), melanocytes (Mel), and adipocytes (Adipo) (Figure 2A). Notably, one fibroblast cluster, FB 4, was expanded after topical treatment with PBMCsec compared with the controls (Figure 2A,B), suggesting an important role in the anti-fibrotic action of PBMCsec. Furthermore, the relative numbers of DCs and TCs were increased with the control medium but slightly reduced with PBMCsec (Figure 2B). We then calculated the differentially expressed genes (DEGs) of all cell populations in PBMCsec-treated scars compared with medium- and NaCl-treated scars. Interestingly, significantly more genes were downregulated than upregulated after the topical application of PBMCsec (Figure 2C), and the highest numbers of regulated genes were found in FBs (red bars), macrophages (pale-green bars), and KCs (yellow bars) (Figure 2C) [35]. To provide an overview of the overall regulation in all cell types, we show the top 50 DEGs per cluster group in Appendix A. The upregulation of numerous genes, previously described to be increased in scar tissue [7,49,50], was significantly inhibited after PBMCsec application.

Next, we analyzed the scRNAsec dataset of scars treated with the intradermal injection of PBMCsec and controls in a similar way. After cluster identification and quality control (Appendix A), clusters aligned homogenously across samples and conditions (Figure 2D). Although the cellular composition of scars did not change after 6 weeks, the FB and immune cell populations were significantly reduced in 8-week-old scars (Figure 2E). Remarkably, there were again far more downregulated genes than upregulated genes in the injected scars, and transcriptome changes were the highest in FB1 and KC clusters (Figure 2F) after injections. Interestingly, only minor transcriptome changes remained in PBMCsec-treated scars 8 weeks after wounding (Figure 2F). The top 50 DEGs after 6 weeks are shown per cluster group in Appendix A. Numerous genes regulated in the topical dataset and previously found relevant in skin scarring and mouse scar formation [7] were also regulated after PBMCsec injection (Appendix A).

As the highest number of regulated genes was observed in FBs and FBs are the main cell type involved in fibrotic processes, we further performed a gene ontology analysis of genes downregulated by PBMCsec application in FBs in both experimental settings (Figure 2G,H).

Our analysis revealed that genes downregulated by PBMCsec mainly showed a strong association with the response to growth factors, integrin activation, monocyte chemotaxis, and extracellular matrix organization, suggesting that the activation of these processes was, at least partially, reduced with topical application (Figure 2G). GO term calculation of downregulated genes in FBs after the injection of PBMCsec revealed changes in ECM and collagen organization, the response to growth factor stimulus, and Wnt signaling (Figure 2H).

Taken together, these bioinformatic data suggest an anti-fibrotic, anti-inflammatory effect of PBMCsec on scar formation, primarily reducing excessive matrix deposition.

### 3.3. PBMCsec Significantly Alters the Matrisome

Since FBs contributed the most to transcriptome alterations induced by PBMCsec and the GO analysis indicated that genes associated with the ECM were highly affected, we further assessed genes of the matrisome in more detail. Differentially regulated genes in all FBs after topical (Figure 3A–D) and intradermal injection (Figure 3E–H) were analyzed using the curated matrisome gene set enrichment analysis (GSEA) gene lists [51]. For better visualization, the whole matrisome was split into the main components, i.e., collagens, proteoglycans, glycoproteins, and ECM regulators. Interestingly, most of the matrisome-related genes were strongly downregulated by PBMCsec after topical and intradermal application (Figure 3). Similarly, most of the proteoglycans, glycoproteins, and ECM regulators showed reduced expression after PBMCsec treatment. However, some of the glycoproteins and ECM regulators, including *Fn1*, *Igfbp4/5*, *Ecm1*, *Postn*, and *Mfap5*, were even enhanced after PBMCsec treatment (Figure 3C,D), suggesting the targeted regulation of these factors.

Importantly, we also identified a variety of proteases, including *Mmp19* (matrix metalloprotease 19), *Ppcsk5/6* (Subtilisin/Kexin-Like Protease PC5/6), and *Adamts1* (A disintegrin-like and metallopeptidase with thrombospondin type 1 motif), regulated by PBMCsec. Furthermore, plasminogen activator/urokinase (*Plau*) and the plasminogen activator/tissue type (*Plat*), as well as serine proteases *Htra1*, *Htra3*, and *Aebp1*, were elevated after the topical application and intradermal injection of PBMCsec. However, a variety of protease inhibitors, including *Timp1* and *-3* (Metallopeptidase Inhibitor 1 and 3), and *Slpi* (Secretory Leukocyte Protease Inhibitor), and the potent urokinase inhibitors *Serpine1, Serpinb2*, and *Serpinb5* were also increased (Figure 3C,D). These findings confirm our previous work, highlighting the role of proteases and their inhibitors in skin fibrosis [7], and indicate that PBMCsec is able to interfere with the protease system that contributes to scar formation. 

### 3.4. Scars Treated with PBMCsec Ex Vivo Show Strong Similarities to Mouse Models

As we showed an anti-fibrotic effect of PBMCsec during scar formation in mice, we next investigated its effect on human skin and ex vivo cultures of scar tissue. Therefore, we treated biopsies of human skin and human hypertrophic scars with medium or PBMCsec and cultivated them for 24 h (Figure 4A). After quality control and cluster identification (Appendix A), clusters aligned homogeneously across donors and conditions (Figure 4B and Appendix A). As described in our previous work, the ratio of FBs was increased in scars compared with skin [7], and several FB clusters (here, clusters FB5 and FB7) were specifically found in scars (Figure 4B,C). Remarkably, the percentages of FBs, DCs, and T cells were reduced in scars after PBMCsec treatment (Figure 4C).

Next, we calculated DEGs separately for skin (Appendix A) and scars (Appendix A) and found a much higher number of DEGs in scars than in normal skin, indicating a strong effect of PBMCsec on fibrotic tissue (Figure 4D). In line with our mouse datasets, most regulated genes were found in the FB clusters, and slightly more genes were downregulated than upregulated, particularly in skin tissue (Figure 4D). Numerous genes that we previously described for their regulation in hypertrophic scars [7] were also favorably regulated by PBMCsec (Appendix A).

Next, we performed the GO term analysis of the DEGs in FBs treated with PBMCsec compared with medium. In line with the mouse data, downregulated terms (Figure 4F) included collagen fibril and ECM organization, cytokine signaling pathway, negative regulation of signal transduction, regulation of extrinsic apoptotic signaling pathway, and type I interferon signaling pathway. Intriguingly, among the upregulated terms (Figure 4E), negative regulation of neuron differentiation and generation of neurons were present. As we previously demonstrated that Schwann cells promote ECM formation in keloids and affect the M2 polarization of macrophages [52], this finding might hint at a mechanism of PBMCsec also affecting this crosstalk.

Next, we assessed the genes of the matrisome in the human dataset (Figure 4H). Similarly to the data obtained for mouse scars, collagens *COL1A1*, *COL3A1*, and *COL6A1/2/3* were also strongly downregulated, more in scars than in skin, and proteases *MMP1/MMP3/10* as well as protease inhibitors *SERPINE1/G1/F1/B2*, *SLPI*, and *TIMP3* were upregulated (Figure 4D). Of note, PBMCsec increased the expression of *PI3*, an elastase-specific protease inhibitor in human scar tissue, indicating a regulatory effect not only on collagens but also on elastic ECM components. Together, our analysis of human ex vivo skin and scars corroborated the findings of the in vivo mouse experiments, indicating an ECM-balancing, anti-fibrotic effect.

### 3.5. PBMCsec Abolishes Myofibroblast Differentiation In Vitro

After a comprehensive analysis of the effects of PBMCsec in mouse and human models at the single-cell level, we investigated the underlying mechanisms of the observed anti-fibrotic activity in vitro. Using a well-established in vitro fibrosis model [7,53], we stimulated primary human skin FBs with TGFβ1 and investigated the effect of PBMCsec on myofibroblast (myoFB) formation [54]. Upon the stimulation of FBs with TGFβ1, FBs showed robust differentiation to αSMA-expressing myoFBs in all control treatments (NaCl and medium) (Figure 5A). In contrast, the addition of PBMCsec completely abolished myoFB differentiation and αSMA expression (Figure 5A,B). As our scRNAseq revealed that of all major ECM components, *Eln*/*ELN* was the most consistently downregulated one in the matrisome of both mice and humans, we further assessed the effect of PBMCsec on the expression of elastin in vitro in FBs. Strikingly, elastin protein and mRNA expression were strongly downregulated by PBMCsec (Figure 5A,C), and the secretion of ELN in the supernatant was significantly inhibited (Figure 5D). Next, we investigated whether PBMCsec contains TGFβ inhibitors. Therefore, we used an HEK-cell-based reporter assay to assess the activity of canonical TGFβ1 signaling. While PBMCsec showed little-to-no TGFβ1 activity, the addition of PBMCsec to active TGFβ1 did not inhibit canonical TGFβ1 activity (Appendix A). These data indicate that PBMCsec does not inhibit myoFB differentiation by inhibiting Smad2/3-mediated TGFβ1 activity, suggesting a more downstream inhibitory or non-canonical action.

To confirm the observed TGFβ effects in vivo, we injected TGFβ1 into murine skin (modified after Thielitz et al. [53]) for 5 consecutive days (Appendix A). Although no morphological changes were visible in hematoxylin–eosin staining (Appendix A), the immunostaining of Collagen I and III showed patches of increased matrix deposition in all samples (arrows in Appendix A), which were not present in mice also treated with PBMCsec. Remarkably, we also observed accumulations of αSMA-expressing cells in the TGFβ1-injected deep murine dermis (squares), but not in PBMCsec-treated mice (Appendix A).

Next, we aimed to further investigate changes in ECM composition, particularly elastin, in a human model. Thus, we injected TGFβ intradermally in human skin explants with and without NaCl, medium, or PBMCsec (Figure 5E). Morphologically, no changes were observed in H&E staining (Figure 5F); however, when we stained for overall ECM configuration using Elastica van Giesson staining (Figure 5G) and with immunofluorescence for elastin (Figure 5H), we noticed specific subepidermal alterations in elastic fibers. In untreated skin, elastin showed vertical fibers reaching into the dermal papillae with parallel, horizontal fibers in the deeper dermis. These vertical, papillary fibers disappeared after TGFβ1 treatment but were preserved when PBMCsec was added (Figure 5G,H). These data suggest that PBMCsec is able to reduce the breakdown of elastic fibers, which occurs after TGFβ stimulation.

### 3.6. Combined Analysis of Murine and Human scRNAseq Datasets Reveals Elastin and TXNIP as Joint Key Players of Beneficial PBMCsec Effects

To better understand the mutual mechanisms of action of ECM balancing and anti-fibrotic mechanisms of PBMCsec, we performed the subclustering of the FBs of all scRNAseq datasets (Appendix A) and performed a combined analysis (Figure 6A). As myoFB, i.e., *Acta2*/*ACTA2*-positive FBs, disappear in mature scars [54], these cells were not detected in most of our datasets. Therefore, we were not able to investigate the effects of PBMCsec on myoFB differentiation in our scar models in detail (Appendix A). However, we detected a significant reduction in *ACTA2* in ex vivo PBMCsec-treated human scars (Appendix A), indicating that even in mature scars, PBMCsec can reduce myoFB content. When overlaying DEGs from FBs from all three experiments, no genes were mutually upregulated (Figure 6B). Interestingly, *Eln/ELN* and *Txnip/TXNIP* were mutually downregulated in all experimental settings (Figure 6C). Elastin and TXNIP were solidly reduced in all three scRNAseq, at both time points after injection, and in human scars (Figure 6D,E).

As we have shown that PBMCsec does not interfere with canonical TGFB1 activity, we next wanted to know how TGFβ signaling is inhibited by PBMCsec. TGFβ is one of the most pleiotropic signaling molecules, and its interaction via the regulation of its release and activation by elastin was previously described [55]. TGFβ is secreted as inactive and bound to latent TGFβ binding proteins (LTBP1-4), together forming the large latent complex (LLC) [56]. The activation of TGFβ occurs via a tightly controlled process involving the cleavage of LTBPs or protease-independent activation via integrins [56,57]. We, therefore, wondered whether PBMCsec also regulates molecules indirectly involved in TGFβ activation. Surprisingly, we found that *Ltbp4*/*LTBP4* was decreased by PBMCsec in both mouse and human experimental settings (Figure 6F). *Ltbp4*/*LTBP4* is involved in both elastogenesis and the regulation of TGFβ signaling [57,58], and an increase in Ltbp4 is associated with fibrosis in scleroderma via TGF-β/SMAD signaling [59]. Additionally, we found that the expression of integrin subunits beta 1 and beta 5 (*Itgb*/*ITGB 1*/*5*) was also decreased upon PBMCsec treatment (Figure 6G,H). As both participate in the activation of TGFβ [56], these data indicate that their downregulation might indirectly contribute to the reduction in TGFβ-mediated fibrotic effects.

Finally, we investigated whether PBMCsec contains endogenous elastase inhibitors that inhibit elastin breakdown and the release of TGFβ [60], further enhancing the anti-TGFβ feedback loop induced by PBMCsec. However, the elastase activity assay showed only a weak reduction in elastase activity after the addition of PBMCsec (Figure 6H). We, therefore, propose a multi-effect model for the attenuation of fibrosis with PBMCsec (Figure 6J): PBMCsec directly inhibits TGFβ1-mediated myoFB differentiation, but not via canonical signaling. PBMCsec attenuates the expression of numerous matrix genes and significantly reduces elastin secretion. PBMCsec prevents elastin breakdown, shows mild elastase inhibition, and interferes with TGFβ-induced gene expression (Figure 6J).

## 4. Discussion

For patients, scars, particularly hypertrophic scars, not only represent an aesthetic problem but often lead to significantly reduced quality of life due to associated limitations of movement, itching, and pain [8]. As the treatment of hypertrophic scars remains difficult, the development of new therapeutic options is of particular interest. Here, we present a multi-model approach to assessing the effects of a secretome-based drug (PBMCsec) on scar formation and treatment in mice and humans. The strong tissue-regenerative activity of PBMCsec has already been demonstrated not only in cutaneous wounds [25,26,27] but also in various other organs, such as focal brain ischemia [31], spinal cord injury [32], and infarcted myocardium [33]. Interestingly, in all organs mentioned above, PBMCsec significantly reduced the size of the damaged areas and reduced the developing fibrotic tissue, suggesting its potential use in the treatment of cutaneous scars [27,33], In this study, we compared the effect of PBMCsec on scar formation in mice in vivo and in human ex vivo explant cultures. In mice, we performed the intradermal injection of the secretome into mature scars and applied it topically during wound healing and scar formation. Only a few studies have investigated the effects of paracrine factors on cutaneous scarring using cell secretomes from different stem cell types, including umbilical cord stem cells, adipose tissue-derived stem cells, or mesenchymal stem cells [61,62,63]. Arjunan et al. and Liu et al. showed that conditioned medium from umbilical cord Wharton’s jelly stem cells or adipose tissue-derived stem cells reduced the activation and growth of keloidal fibroblasts in in vitro and in vivo keloid models [62]. In addition, Hu et al. suggested a combined treatment of conditioned medium from MSC and botulinum toxin for the treatment of hypertrophic scars [62]. However, in-depth analyses of the underlying mechanisms are still lacking. Thus, our study is the first to use scRNAseq to unravel mechanisms important for improved scar formation after the application of a cell secretome. Generally, scRNAseq generates large datasets with tens of thousands of cells, which helps to smooth out donor and technical variances. Therefore, low donor numbers, as used in our study, are widely acceptable [64,65,66]. 

In our mouse experiments, both application routes, topical and intradermal application, showed promising effects on scar formation and treatment. Of note, significantly more genes were regulated after the topical application of PBMCsec, suggesting higher efficacy after wound application than after injection. However, the improved wound healing process per se after PBMCsec application might already be decisive for better scar quality. Therefore, a direct comparison of the two application routes is difficult and requires further experiments where PBMCsec is topically applied to already existing scars. Furthermore, other potential treatment options, such as application after laser treatment [67], microneedling [68,69], or in combination with nanocarriers [68] should be tested in future experiments. Most importantly, and in line with the data on mouse scar formation, we also identified a significant anti-fibrotic effect of PBMCsec on human mature hypertrophic scars in explant cultures. In fact, the treatment of scars with PBMCsec in mice and humans showed high similarities. In both species, we found the strongest transcriptome alterations in FB clusters, specifically in genes of the so-called matrisome, which includes collagens, proteoglycans, glycoproteins, and ECM regulators [22,23,24]. The matrisome, which was recently defined for large-scale in silico analyses, provides a comprehensive overview of the components of the ECM [51,70,71]. Although several characteristics of ECM alterations in (hypertrophic) scars have already been described [72], our study provides the first large dataset analyzing changes in the entire matrisome in mice and humans during wound healing and scar formation. These highly valuable datasets could be the basis for many future studies on the pathophysiology of wound healing and scar formation, as well as on the effects of secretome-based scar treatment.

In the present study, we further focused on elastin, which was similarly downregulated by PBMCsec under all conditions and in all species investigated. Elastin fibril sequences interact with microfibrils and bind to cell surface receptors [73]. Elastin is extremely durable and has a half-life of ~70 years [73,74]. While intact elastin is inert and insoluble, it can be degraded by a plethora of elastases [74], including MMPs, aspartic proteases, serine proteases, and cysteine proteases [74]. In our ex vivo assays, we found strong degradation of elastic fibers in human skin induced by TGFβ, which was completely inhibited by PBMCsec, suggesting an elastase-inhibiting effect of PBMCsec. Intriguingly, this effect of TGFβ on elastic fibers appears to be counterintuitive, and we did not find any other study describing this phenomenon. The interaction of TGFβ and elastin is complex. TGFβ is generally known to induce elastogenesis [47]), stabilize elastin mRNA [47,48]), and increase elastin secretion (Figure 5), which is most likely due to the post-transcriptional control of elastin [47]. This is in line with our in vitro findings, as we could show the strong upregulation of elastin production in fibroblasts treated with TGFβ. Interestingly, this upregulation was also significantly inhibited by PBMCsec at the mRNA and protein level. So far, we cannot offer an explanation for this phenomenon. It is tempting to speculate that the proteolytic breakdown of elastin triggers the de novo synthesis of elastin. Furthermore, whether the TGFβ-induced overproduction of elastin also leads to the assembly of new functional elastic fibers is still not fully understood. Therefore, the mechanisms by which PBMCsec inhibits elastin breakdown need further investigations. Interestingly, our in vitro elastase assay showed only weak anti-elastase activity of PBMCsec, suggesting that either the specific enzyme inhibited by PBMCsec is not detected by the in vitro assay or PBMCsec leads to the induction of endogenous protease inhibitors. In line with the second hypothesis, Copic et al. recently showed that PBMCsec is indeed able to induce the production of SERPINB2, a serine protease inhibitor, in human mononuclear cells [75]. Furthermore, with scRNAseq, we showed that some elastase inhibitors, such as *PI3* (peptidase inhibitor 3) and *SLPI* (secretory leukocyte protease inhibitor), were significantly upregulated by PBMCsec in FBs in scars (Figure 4H). Despite having been well investigated for their beneficial effects in cystic fibrosis [76], these elastase inhibitors have been hardly assessed for their role in cutaneous scar formation so far. Further, more sophisticated experiments are needed to fully address the role of these enzyme inhibitors in scar formation.

Aside from elastin, the only other gene consistently regulated by PBMCsec in all three scRNAseq experimental approaches was *TXNIP* (Thioredoxin interacting protein). TXNIP is critically involved in the regulation of reactive oxygen species (ROS) and cellular oxidative stress [77] and was shown to contribute to disturbed wound healing under ischemic conditions [78]. With regard to scar formation, TXNIP was shown to be elevated in a murine model of pulmonary fibrosis, and the inhibition of TXNIP in this model led to the reduction in ROS and myoFB differentiation [79]. The exact role of TXNIP in skin pathologies and in scars, however, has been scarcely investigated [80]. Our finding that the downregulation of *TXNIP* was conserved across all our experimental approaches suggests that PBMCsec-induced *TXNIP* downregulation might be an important mechanism contributing to the anti-fibrotic action of PBMCsec. However, further studies are needed to fully decipher the mechanism of TXNIP-regulation as well as its impact on cutaneous scar formation.

Interestingly, PBMCsec also prevented FB activation and myoFB differentiation. In line with our results, previous studies showed that treatment of FBs with conditioned medium of mesenchymal or pluripotent stem cells was able to reduce myoFB differentiation [81,82]. In contrast to these studies, we were not able to identify a direct inhibitory action of PBMCsec on canonical TGFβ/Smad signaling [82]. However, TGFβ has been shown to also induce fibrosis via non-canonical (non-SMAD) signaling pathways [83], and blocking non-canonical signaling prevents pro-fibrotic phenotypes [84]. Possible non-canonical pathways might include glycogen synthase kinase-3β (GSK-3β) [85], a pathway we previously found to be regulated upon non-SMAD TGFβ-mediated abolishment of myoFB differentiation [7]. Hitherto, only few secreted molecules inhibiting non-canonical TGF-signaling have been described. Del-1 (Developmentally-Regulated Endothelial Cell Locus 1 Protein) was shown to inhibit TGFβ and attenuate fibrosis by suppressing the α_v_ integrin-mediated activation of TGFβ [86]. In addition, several proteins, such as fibroblast growth factor (FGF), epidermal growth factor (IGF), interferon gamma, and IL-10, all of which are present in PBMCsec, are known to inhibit myoFB differentiation [6]. To identify the exact pathway of TGFβ inhibition induced by PBMCsec, a detailed proteomic approach and the assessment of multiple pathways will be necessary in the future.

As previously discussed [7], there are some limitations to the current study that need to be considered. There are significant differences between the wound healing mechanisms of mice and humans. While mice mainly rely on the contraction of the subcutaneous panniculus carnosus, human wound healing is characterized by the deposition of extracellular matrix (ECM) followed by re-epithelialization [86,87]. However, recent research has shown that both processes contribute to a similar extent in mice [88]. Therefore, mouse wound models may be considered a valid model for human wound healing. However, it is important to note that the current mouse models of scarring do not fully replicate the pathological fibrotic state observed in human hypertrophic scars. Although mouse models for hypertrophic scars have been developed, such as subcutaneous bleomycin injection [89] and tight-skin mice [90], the comparability of the transcriptome of these models with human hypertrophic scars is not yet fully understood.

In conclusion, we provide an extensive study with multiple experimental approaches and ample scRNAseq data. Comprehensive analyses suggest a solid anti-fibrotic, ECM reducing, and myoFB-inhibiting effect of PBMCsec. We identified the prevention of elastin breakdown as a putative major underlying mechanism of PBMCsec-mediated scar attenuation. We thus propose future clinical assessment of PBMCsec to attenuate skin scarring during wound healing and to treat already existing mature scars [37].

## Figures and Tables

**Figure 1 pharmaceutics-15-01065-f001:**
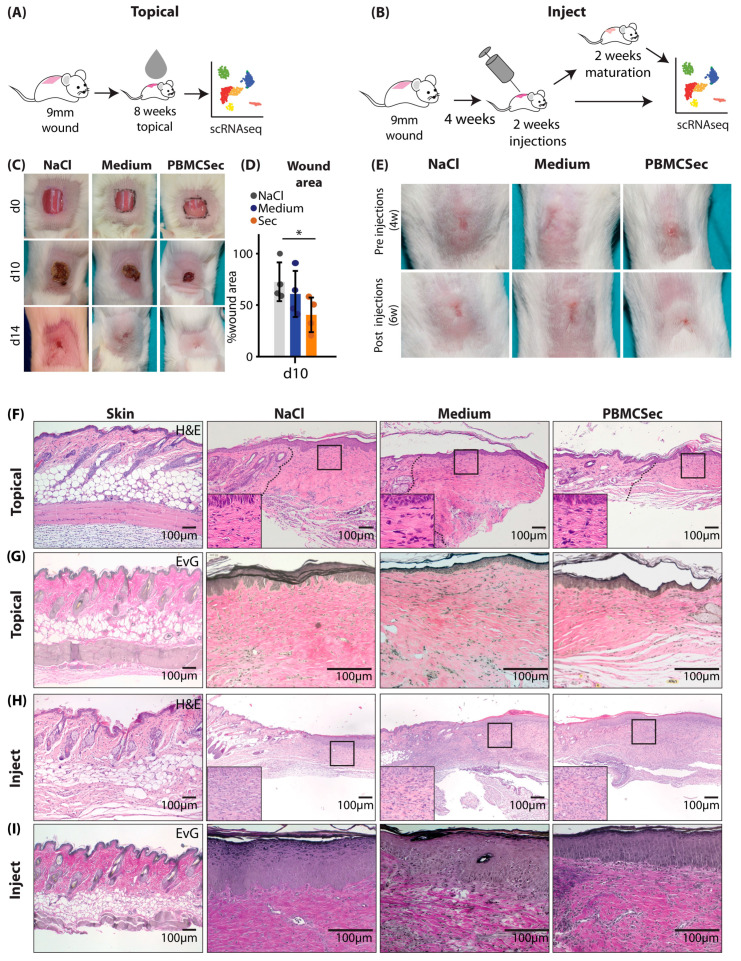
Comparison of PBMCsec-mediated effects on scars after topical or intradermal application. (**A**,**B**) Illustration of “topical” or “inject” workflow for mouse scars: (**A**) square wounds of 9 × 9 mm were excised on mouse backs (n = 4 per group), treated with topical PBMCsec for 8 weeks, and subjected to scRNAseq; (B) square wounds of 9 × 9 mm were excised on mouse backs (n = 4 per group), left to mature for 4 weeks, injected with PBMCsec for 2 weeks, and subjected to scRNAseq, or matured for another 2 weeks and then subjected to scRNAseq. (**C**) Wound documentation of topically treated scars on post-wounding days 0, 10, and 14. * indicates *p* < 0.05 in one-way ANOVA (**D**) Wound area measurements normalized to the d0 wound area of each respective wound. (**E**) Scar documentation of mouse scars before and after injections. (**F**) Hematoxylin/eosin staining of ”topical” mouse wounds. (**G**) Elastica van Gieson (EvG) staining of ”topical” mouse wounds. (**H**) Hematoxylin/eosin staining of ”inject” mouse wounds. (**I**) Elastica van Gieson (EvG) staining of ”inject” mouse wounds.

**Figure 2 pharmaceutics-15-01065-f002:**
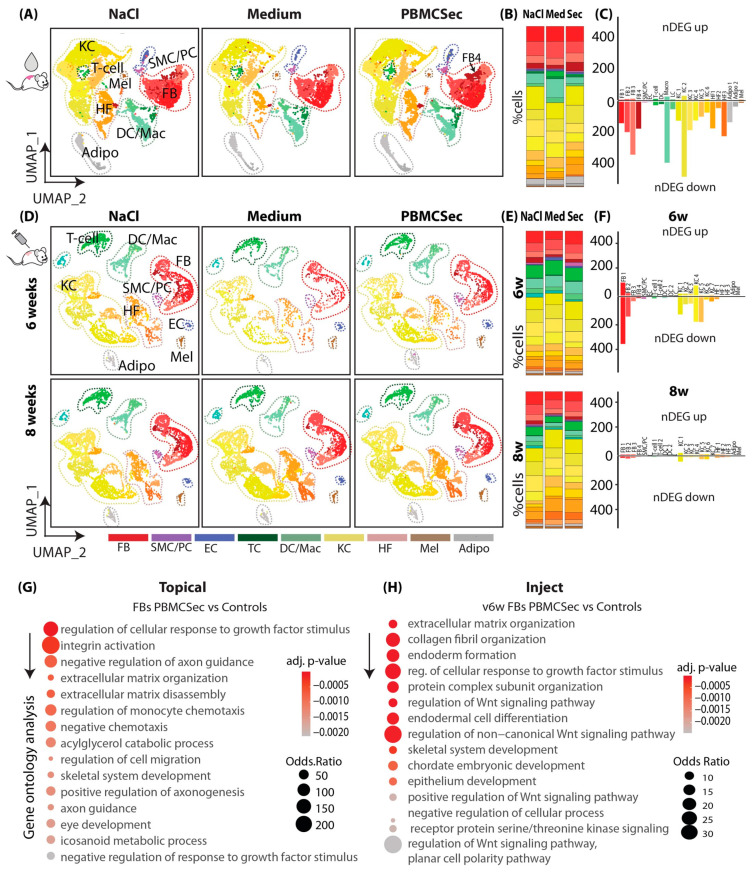
PBMCsec induces significant changes in the transcriptome after topical and intradermal application. (**A**) UMAP clustering of ”topical” mouse wounds (n = 4 per condition, pooled for scRNAseq analysis), split by condition: four fibroblast clusters (FB1-4; red), smooth muscle cells (SMCs) and pericytes (PCs; purple), endothelial cells (ECs; blue), T cells (TCs; dark green), macrophages (Mac) and dendritic cells (DCs; light green), three keratinocyte clusters (KC1-6; yellow), hair follicles (HF 1-3; beige), melanocytes (Mel; brown), and adipocytes (grey). Clusters were grouped as “FB”, “PC”, “TC”, “DC”, “KC”, “HF”, “MEL”, and “Adipo” for readability. (**B**) Percentages of cells per cluster, split by condition. (**C**) Number of significantly upregulated (positive y-axis) and downregulated (negative y-axis) genes (“nDEG”) per cluster in “topical” mice. (**D**) UMAP clustering of “inject” mouse wounds (n = 2 per condition), split by condition, i.e., 6w = mice after two weeks of injections; 8w = mice after injections + 2 weeks of maturation. Clusters FB1–4, SMCs, PCs, ECs, T cells 1+2, DC1+2, KC1–7, HF 1–3, Mel, and Adipo. Clusters were grouped as “FB” (red), “PC” (purple), “EC” (blue), “TC” (dark green), “DC” (light green), “KC” (yellow), “HF” (beige), “MEL” (brown), and “Adipo” (grey) for readability. (**E**) Percentages of cells per cluster, split by condition. (**F**) Number of significantly upregulated (positive y-axis) and downregulated (negative y-axis) genes (“nDEG”) per cluster in “inject” mice, split in 6w and 8w (**G**) Gene ontology (GO) term calculation of genes downregulated by PBMCsec compared with medium in “topical” FBs. (**H**) GO term calculation of genes downregulated by PBMCsec vs. medium in 6w “inject” FBs. DEGs were calculated per cluster comparing 8- and 6-week-old scars using a two-sided Wilcoxon-signed rank test, including genes with average logarithmic fold change (avg_logFC) of >0.1 or <−0.1.; adj. *p*-value < 0.05. UMAP, uniform manifold approximation and projection.

**Figure 3 pharmaceutics-15-01065-f003:**
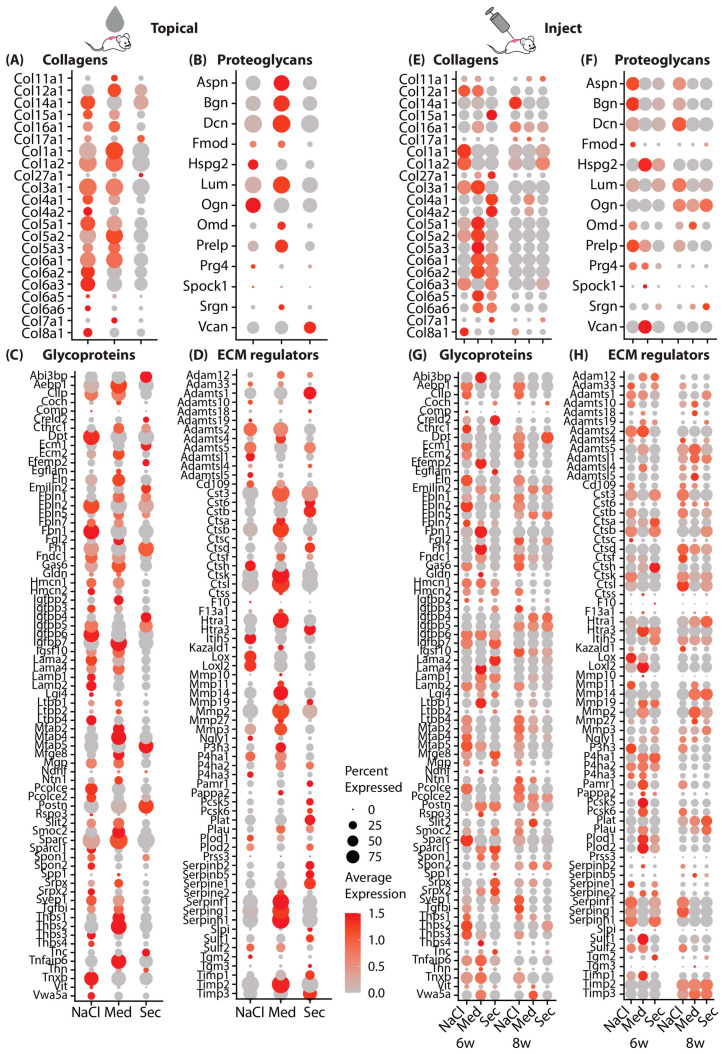
PBMCsec significantly alters the matrisome. Dot plots of gene lists of gene set enrichment of matrisome terms (**A**,**E**) “collagens”, (**B**,**F**) “proteoglycans”, (**C**,**G**) “Glycoproteins”, and (**D**,**H**) “ECM regulators” inputted to FBs of the “topical” (**A**–**D**) and “inject” (**E**–**H**) datasets, split by condition. Circle size correlates with the percent of cells expressing the respective gene, and color (red) correlates with normalized fold change in expression.

**Figure 4 pharmaceutics-15-01065-f004:**
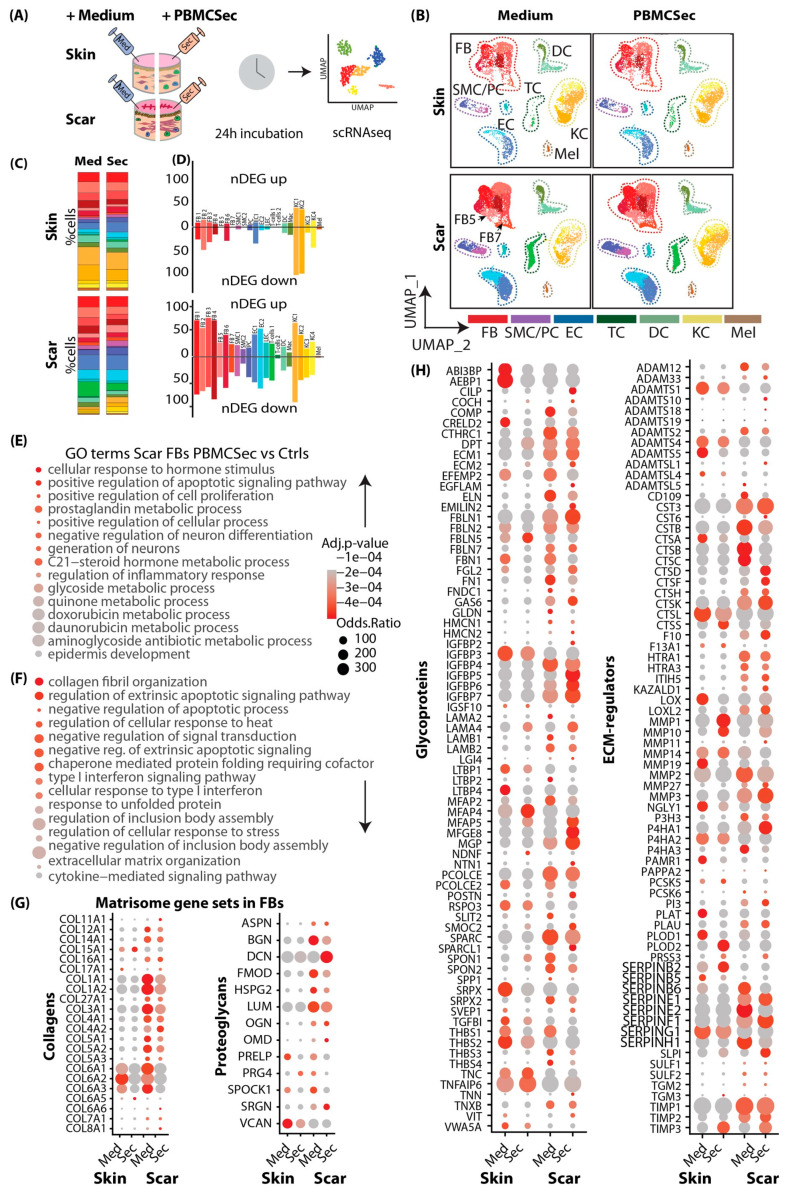
scRNAseq analysis of human skin and scars treated with PBMCsec ex vivo shows strong similarities to mouse models. (**A**) Illustration of scRNAseq workflow in human skin and scar samples. Skin and scar biopsies were incubated overnight in medium or PBMCsec and subjected to scRNAseq. (**B**) UMAP clustering of human skin and scar, split by condition. Seven fibroblast clusters (FB 1–7; red), smooth muscle cells (SMCs) and pericytes (PCs; purple), endothelial cells (ECs; blue), T cells (TCs; dark green), macrophages (Mac) and dendritic cells (DCs; light green), four keratinocyte clusters (KC1–4; yellow), and melanocytes (Mel; brown). Clusters were grouped as “FB”, “PC”, “TC”, “DC”, “KC”, “MEL”, and “HF” for readability. (**C**) Percentages of cells per cluster, split by condition. (**D**) Number of significantly upregulated (positive y-axis) and downregulated (negative y-axis) genes (“nDEG”). (**E**) Gene ontology (GO) term calculation of genes (**E**) upregulated and (**F**) downregulated by PBMCsec compared with medium in “topical” FBs. (**G**) Dot plots of gene lists of gene set enrichment of matrisome terms “collagens” and “proteoglycans”, and (**H**) “Glycoproteins” and “ECM-regulators” inputted in FBs. DEGs were calculated per cluster by comparing 8- vs. 6-week-old scars using a two-sided Wilcoxon signed-rank test, including genes with average logarithmic fold change (avg_logFC) of >0.1 or <−0.1.; adj. *p*-value < 0.05. UMAP, uniform manifold approximation and projection.

**Figure 5 pharmaceutics-15-01065-f005:**
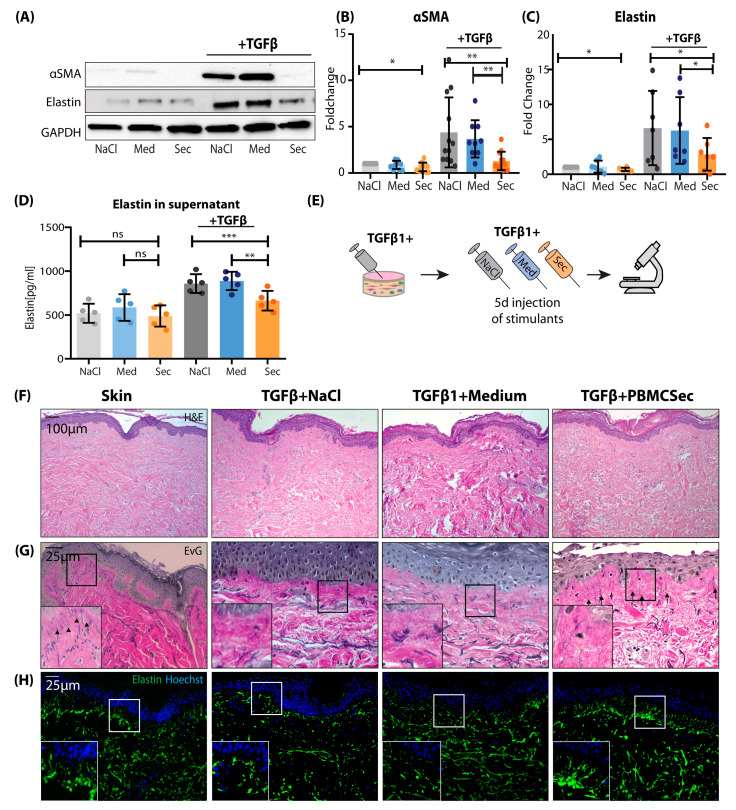
PBMCsec abolishes myofibroblast differentiation in vitro. (**A**) Western blot stained for alpha Smooth muscle actin (SMA) and elastin, and lysate from human primary FBs stimulated with NaCl, medium, or PBMCsec, without or with TGFß1, respectively. (**B**) Quantification of Western blot, normalized to ctrl (n = 6 human donors). (**C**) Elastin measured with ELISA from primary human FB supernatant, stimulated with NaCl, medium, or PBMCsec, without or with TGFß1. (**D**) Workflow illustration of ex vivo human skin TGFß stimulation experiment, where 5 mm skin biopsies were injected with TGFß and NaCl, medium, or PBMCsec for 5 consecutive days (**E**). (**E**,**F**,**H**) Elastica van Gieson. (**G**) Immunofluoresence staining for elastin in human ex vivo skin samples. Statistical significance was tested using one-sided ANOVA. Lines and error bars indicate means and standard deviation. ns *p* > 0.05, * *p* < 0.05, ** *p* < 0.01, and *** *p* < 0.001.

**Figure 6 pharmaceutics-15-01065-f006:**
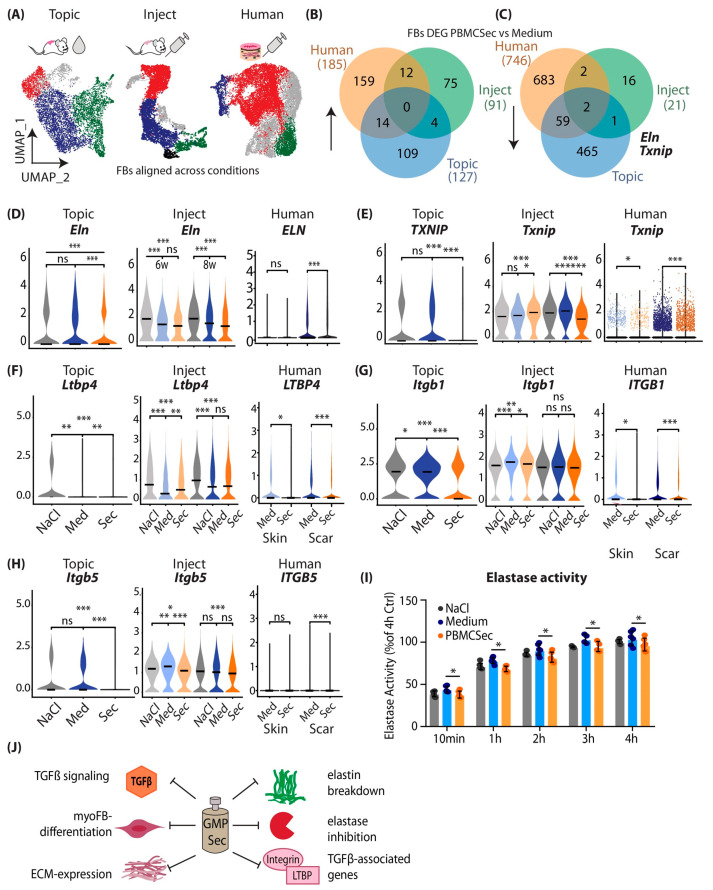
Combined analysis of murine and human scRNAseq datasets reveals elastin as joint key player of beneficial PBMCsec effects. (**A**) Subclustering of FBs in “mouse topical”, “mouse inject”, and “human” scRNAseq datasets, and FB subcluster alignment. Red, cluster A; blue, cluster B; green, cluster C; aligned by cluster markers (Appendix A). (**B**) Venn diagram of overlap of significantly upregulated and (**C**) downregulated genes in FBs in all three datasets. (**D**–**G**) Violin plots of (**F**) latent TGFβ binding protein 4 (Ltbp4/LTBP4) and (**G**,**H**) Integrin subunit beta 1/5 (Itgb1/5/ITGB1/5) in datasets. (**I**) Elastase assay with fluorescence-marked pig pancreas elastase, with NaCl, medium, or PBMCsec supplementation. Y-axis indicates fluorescence intensity, i.e., elastase activity. Comparison among groups was performed with Student’s *t*-test. (**J**) Illustration of putative mechanisms of PBMCsec in scars. In violin plots, dots represent individual cells; y-axis represents log2 fold change in normalized genes and log-transformed single-cell expression. Vertical lines in violin plots represent maximum expression; the shape of each violin represents all results; and the width of each violin represents the frequency of cells at the respective expression level. DEGs were calculated in FBs by comparing medium to FBs using a two-sided Wilcoxon signed-rank test, including genes with average logarithmic fold change (avg_logFC) of >0.1 or <−0.1 and Bonferroni-adjusted *p*-value < 0.05. For violin plots, a two-sided Wilcoxon signed-rank test was used in R. ns *p* > 0.05, * *p* > 0.05, ** *p* > 0.01, and *** *p* > 0.001.

## Data Availability

The scRNAseq data generated in this study have been deposited in the NCBI GEO database under accession numbers GSE156326 and GSE202544. The raw sequencing data are protected and are not available due to data privacy laws. If raw sequencing data are absolutely necessary for the replication or extension of our research, they will be made available upon request to the corresponding author in a 2-week timeframe. All other relevant data supporting the key findings of this study are available within the article and its Supplementary Information files or from the corresponding author upon reasonable request.

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
