# Peer review of "The Secretome of Irradiated Peripheral Mononuclear Cells Attenuates Hypertrophic Skin Scarring"

_pharmaceutics, 2023, doi:10.3390/pharmaceutics15041065_

Round 1

Reviewer 1 Report

This work investigated the effects of secreted factors from peripheral blood mononuclear cells (PBMCsec) on skin scarring in mouse models and human scar explant cultures at single cell resolution (scRNAseq). Mouse wounds and scars and human mature scars were treated with PBMCsec intradermally and topically. The results are detailed, but the following questions need to be addresssed before being accepted.

1. The quality of the pictures needs to be improved.

2. Where is Figure 2D?

3. There appear to be a number of grammatical errors/typos throughout the manuscript, which would benefit from an additional proof-reading. For example: Line 107 "0,1 ml/10mg, 7,5 mg/ml", Line 116"25×106 cells/mL"...

Author Response

Reply to Reviewer #1

Comment 1: The quality of the pictures needs to be improved.

Reply to comment 1: We apologize for the low quality of the figures. We have now included all images in the highest possible resolution.

Comment 2: Where is Figure 2D?

Reply to comment 2: We thank the reviewer for pointing out this error. The figure was renumbered accordingly.

Comment 3: There appear to be a number of grammatical errors/typos throughout the manuscript, which would benefit from an additional proofreading.

Reply to comment 3: As suggested by the reviewer we have performed additional proofreading.

Reviewer 2 Report

The work is very interesting and well done, there are only some inaccuracies to review:

Line 30: insert a period and not a comma at the end of the keywords

Line 137: When you indicate 4 parts ULTRASICC and ULTRABAS may you specify the ratio between the two base creams, one being O/W and the other W/O?

Lines 85-172 and 226-238 and 248-262: the text is not justified

Lines 175-198; 207-211 and 220-224: the text has a different size, harmonize it

Author Response

Reply to Reviewer #2:

The work is very interesting and well done, there are only some inaccuracies to review:

Comment 1: Line 30: insert a period and not a comma at the end of the keywords

Reply to comment 1: This error has been corrected

Comment 2: Line 137: When you indicate 4 parts ULTRASICC and ULTRABAS may you specify the ratio between the two base creams, one being O/W and the other W/O?

Reply to comment 2: Equal amounts of Ultrasicc and Ultrabas were mixed together to obtain a 50:50 mixture. This information is now included in the revised manuscript.

Comment 3: Lines 85-172 and 226-238 and 248-262: the text is not justified

Reply to comment 3: The indicated lines are part of our Materials and Methods section. specifically, lines 85-172 consist of our ethics statement, the description of the patient material, animals used in the study, our murine wound and scar model, the production of our secretome, the protocol of secretome injection and topical application, ex vivo skin experiments and parts of the single cell analysis. Furthermore, lines 226-238 contain the description of the used Western blot and staining procedures, and lines 248-262 describe the methodology and used reagents for the elastase and ELISA assays.

All of these Materials and Methods are part of the manuscript and must be described. Therefore, we have decided to leave these paragraphs in the revised version of our manuscript.

Comment 4: Lines 175-198; 207-211 and 220-224: the text has a different size, harmonize it

Reply to comment 4: This error has been corrected.

Reviewer 3 Report

The authors have used the product of irradiated mononuclear cells to reduce hypertrophic scarring. There were some problems with the manuscript:

1) In the mouse model, I did not see much in the way of differences with treatment. Mice are a very poor model for examining scarring since they contract their wounds so fast. Your treatment lasted for 8 weeks but the animals must have been healed within 2-3 weeks. In addition, you use a very low number of mice (2 in some experiments), can you reproduce the results?

2) Monocytes produce scores of proteins. If their product does affect scarring are you going to find out what is causing the effect?

3) Can you comment on how how the expression of cultured fibroblasts actually relate to human scarring?

4) Do you have a plan on how to use the product in actual human scarring?

Author Response

Reply to Reviewer #3:

Comment 1: In the mouse model, I did not see much in the way of differences with treatment.

Reply to comment 1: We agree with the reviewer that the macroscopic and histological differences are subtle, and that conclusions should not be solely drawn from the photographic documentation. However, our scRNAseq analysis showed significant differences, and in conjunction with the observed differences in experiments with human material, we are confident in the description of our findings and the conclusions drawn.

Comment 2. Mice are a very poor model for examining scarring since they contract their wounds so fast.

Reply to comment 2: The reviewer has raised an important point, which we have extensively discussed and studied in our previous work (Vorstandlechner et al., doi: 10.1038/s41467-021-26495-2). There are certainly considerable differences between human and murine wound healing; While mouse wounds heal predominantly via contraction promoted by the subcutaneous panniculus carnosus, de novo formation and deposition of ECM and subsequent re-epithelialization prevail in human wound healing. However, a study assessing the contribution of epithelialization and contraction in mice found that each accounted for 40–60%, and that mouse wound models can thus be considered a valid model for human wound healing (Chen, L., Mirza, R., Kwon, Y., DiPietro, L. A. & Koh, T. J. The murine excisional wound model: contraction revisited. Wound Repair Regen. 23, 874–877 (2015)). Moreover, our mouse scarring model does not fully reflect the pathological fibrotic state of human hypertrophic scars. Although mouse models for hypertrophic scars, e.g., subcutaneous bleomycin injection, or tight-skin mice have been described, their transcriptome comparability with human hypertrophic scars is not well investigated. However, in combination with our human data we believe that the data from mice are valid and meaningful. We have now extended the discussion of our paper to better address these issues.

Comment 3: Your treatment lasted for 8 weeks but the animals must have been healed within 2-3 weeks. In addition, you use a very low number of mice (2 in some experiments), can you reproduce the results?

Reply to comment 3: The treatment strategy used in this experiment was intended to treat the wounds and also the scars, once wound healing was completed. Therefore, an 8 week treatment strategy was used.

As single-cell sequencing is costly but yields large datasets of tens of thousands of cells, thereby smoothening donor and technical variances, low donor numbers are usually justifiable and eccepted in the scientific community (Mahmoudi, S. et al. Nature 574, 553–558 (2019); Vorstandlechner, V. et al. Nat Comm. 29;12(1):6242 (2021); Solé-Boldo, L. et al. Commun. Biol. 3, 188 (2020); Joost, S. et al. Cell Stem Cell 26, 441–457.e447 (2020)). However, we fully agree that the relatively small sample size in our study should be considered as a limitation of our study.

We have now extended the discussion of our paper to better address these issues.

Comment 4: Monocytes produce scores of proteins. If their product does affect scarring are you going to find out what is causing the effect?

Reply to comment 4: As discussed in this paper and previous studies, our secretome is extremely complex and consists not only of proteins but also lipids, extracellular vesicles, and nucleic acids. We have identified more than 20 mechanisms through which this secretome affects regeneration and tissue remodeling. For example, PBMCsec induces effects that prevent tissue destruction such as platelet inhibition and vasodilation (Hoetzenecker, K. et al. Basic Research in Cardiology 2012) and increased expression of cytoprotective and anti-apoptotic genes in primary cultured human cells. On the other hand, PBMCsec also induces effects that promote tissue regeneration and wound healing such as enhanced migration of fibroblasts and keratinocytes, prevention of dendritic cell and basophil activation  and increased sprouting of aortic and spinal cord endothelial cells in vitro (Laggner, M. et al. EBioMedicine 2022; Laggner, M. et al. EBioMedicine 2020). So far, no single effective component has been determined in mechanism of action studies. Our manuscript highlights that the beneficial effects of PBMCsec results from a synergistic effect of many components of the secretome. Based on the findings of this study, we suggest that PBMCsec improves scarring and regeneration through 1) inhibition of TGFbeta-mediated FB activation and myofibroblasts differentiation; 2) thus reducing the deposition of excessive ECM; 3) inhibition of pro-fibrotic elastase, and 4) thus preserving of elastin fibers and preventing of excessive elastogenesis in scarring. However, as the secretome has already been approved for clinical studies and we are currently running a phase II clinical trial, we refrain from further dissecting the secretome into single components.

Comment 5: Can you comment on how the expression of cultured fibroblasts actually relate to human scarring?

Reply to comment 5: It is well established that treatment of cultured FBs with TGFβ reflects the activation FBs undergo during wound healing and scarring in vivo. Cultured TGFβ-treated primary human skin FBs acquire a contractile phenotype marked by their expression of SMA, along with increased expression of ECM components, making them myofibroblasts. This was well reflected in our human ex vivo experiments, were scRNAseq consistently showed an increase of ECM-expression in scar FBs compared to FBs present in normal skin (Figure 4G). However, in mature human scars, myofibroblasts usually do not persist, but de-differentiate after remodeling. This was also confirmed in our dataset, as no SMA (ACTA2)-positive cells remained in the mature hypertrophic scars (Figure S9 A-C). The in vivo model of myofibroblast differentiation is a widely used and accepted approach to study underlying mechanisms and drugs interfering with this process.

Comment 6: Do you have a plan on how to use the product in actual human scarring?

Reply to comment 6: PBMCsec (APOSEC®) is currently under Phase II investigation for topical application for wound healing of diabetic foot ulcers (Gugerell et al., doi: 10.1186/s13063-020-04948-1). As soon as topical application in wound healing is approved by the Austrian and European authorities, clinical studies for topical application in wounds to prevent (hypertrophic) scarring, most likely with a focus on burnspatients, will be planned. In the future, we plan to perform clinical studies to evaluate intracutaneous injection into already present hypertrophic scars.

Round 2

Reviewer 3 Report

The authors improved the paper. I am not convinced that they showed that their treatment was effective, but I feel that it is worthwhile to accept the paper. 

They are missing the word "to" on line 273. 

Author Response

We thank the editor for the interest in our work and the reviewers for their effort and time put into the thoughtful review and their comments.

Comment 1: The authors improved the paper. I am not convinced that they showed that their treatment was effective, but I feel that it is worthwhile to accept the paper.

Reply to comment 1: We express our gratitude to the reviewer for the thorough evaluation of our manuscript. Our findings indicate that PBMCsec regulates several hundreds of genes in both mouse models, which strongly supports the effectiveness of PBMCsec in these models.

Comment 2: They are missing the word "to" on line 273. 

Reply to comment 2: The missing word in line 273 (which is line 267 in the word file) has been added.